# Quality of Daily-Life Gait: Novel Outcome for Trials that Focus on Balance, Mobility, and Falls

**DOI:** 10.3390/s19204388

**Published:** 2019-10-11

**Authors:** Kimberley S. van Schooten, Mirjam Pijnappels, Stephen R. Lord, Jaap H. van Dieën

**Affiliations:** 1Neuroscience Research Australia, University of New South Wales, Sydney, NSW 2031, Australia; s.lord@neura.edu.au; 2School of Public Health and Community Medicine, Faculty of Medicine, University of New South Wales, Sydney, NSW 2033, Australia; 3Department of Human Movement Sciences, Faculty of Behavioural and Movement Sciences, Vrije Universiteit, Amsterdam Movement Sciences, 1081BT Amsterdam, The Netherlands; m.pijnappels@vu.nl (M.P.); j.van.dieen@vu.nl (J.H.v.D.)

**Keywords:** intervention studies, accelerometry, activity monitoring, aged, accidental falls

## Abstract

Technological advances in inertial sensors allow for monitoring of daily-life gait characteristics as a proxy for fall risk. The quality of daily-life gait could serve as a valuable outcome for intervention trials, but the uptake of these measures relies on their power to detect relevant changes in fall risk. We collected daily-life gait characteristics in 163 older people (aged 77.5 ± 7.5, 107♀) over two measurement weeks that were two weeks apart. We present variance estimates of daily-life gait characteristics that are sensitive to fall risk and estimate the number of participants required to obtain sufficient statistical power for repeated comparisons. The provided data allows for power analyses for studies using daily-life gait quality as outcome. Our results show that the number of participants required (i.e., 8 to 343 depending on the anticipated effect size and between-measurements correlation) is similar to that generally used in fall prevention trials. We propose that the quality of daily-life gait is a promising outcome for intervention studies that focus on improving balance and mobility and reducing falls.

## 1. Introduction

Effective interventions to prevent falls in older people address balance and mobility problems through exercise interventions, medication reviews, and environmental modifications [1]. Despite a focus on balance and mobility, the default primary outcomes of such intervention trials are the rate of falls or the proportion of fallers. These outcomes require 6 to 12 months of intensive follow-up and are susceptible to definitional issues and recall biases [2,3]. Moreover, the incidence of falls does not directly equate fall risk, since exposure to random environmental events plays an important role [4,5]. Clinical tests of balance and mobility can be assessed relatively quickly and can be used as proxies for fall risk in intervention trials. However, their utility has been hampered by the requirement of in-person assessments under standardized conditions, which may not reflect an individual’s performance in daily life [6,7].

Technological advances in wearable sensors allow for the assessment of the quantity and quality of daily-life activities. Several studies have shown that characteristics of daily-life gait, assessed by a single trunk-worn sensor, are associated with fall risk among older people [8,9,10]. These gait characteristics, such as gait stability and variability, provide complementary information to commonly-used clinical tests in the prediction of falls [11]. Experimental studies further provide evidence that gait characteristics, during standardized assessments, are sensitive to relatively small balance impairments and are affected by disturbances of the sensory systems, medication, muscle fatigue, and training [12,13,14,15,16]. Effect sizes for these manipulations approximate a Cohen’s *d* of 3.0–5.0 for desensitization of the feet and electrical vestibular stimulation in healthy young [13,14], 0.3 for rivastigmine medication in people with Parkinson’s disease [16], 0.5–0.6 for muscle fatigue in older people [12] and 0.3–0.8 for a dance intervention in older people [15]. The difference in daily-life gait quality between older people who fall, and those who do not fall in the next 6 months, is in the order of a Cohen’s *d* of 0.3 (range 0.03 to 0.5) [11]. The assessment of daily-life gait characteristics may hence be valuable for intervention trials. 

In order to determine whether daily-life gait characteristics are sensitive to change, such as intervention effects, large-scale clinical trials are required. Sample size calculations are essential in the design of such trials. However, the estimation of the variance components that are needed for these calculations, requires large samples of repeated measurements, which are generally not available early in a study. We analyzed trunk accelerometry data collected for [10] to provide this information to support the design of intervention trials. We estimated the variance components and performed a sample size calculation to reveal how many participants would be required in order to detect the statistically significant intervention effects on daily-life gait quality characteristics obtained from inertial sensor data in older people. 

## 2. Materials and Methods

### 2.1. Participants

This study was part of a larger project investigating fall risk in older people (FARAO [8,10]). The participants were older people, who were recruited from Amsterdam (the Netherlands) and its surroundings via advertisements through general practitioners, hospitals, and residential care facilities. Inclusion criteria were: being between 65 and 99 years of age, having no major cognitive impairments (assessed as a Mini Mental State Examination score exceeding 18 out of 30 points [17]), and being able to walk at least 20 m with, or without, the help of a walking aid. The medical ethics committee of the VU medical centre approved the protocol (ID 2010/290) and all participants signed informed consent. 

### 2.2. Measurements

The data of 163 older people (mean age 77.5, SD 7.5 years; 107♀) were analyzed for the current paper. The participants wore a trunk accelerometer (DynaPort MoveMonitor, McRoberts BV, The Netherlands) for two separate one-week periods, with a median between-assessment interval of 14 days (interquartile range 28 days, maximum 65 days). They were instructed to wear the accelerometer at all times, except during water activities, such as showering or swimming as this would damage the device. No intervention took place during the between-assessment interval. The tri-axial trunk accelerometer was placed on the back of the trunk at the level of L5 using a supplied elastic belt, and registered trunk accelerations in vertical (VT), mediolateral (ML), and anteroposterior (AP) directions with a sample frequency of 100 samples/s and range of +/-6 *g*. 

### 2.3. Data Analysis

The episodes of locomotion were identified using the manufacturers algorithm. This algorithm was validated against video [18] and the detection of locomotion episodes was shown to be reliable over weeks [19]. The raw data of locomotion periods were extracted and realigned with anatomical axes, based on the accelerometer’s orientation with respect to the gravity and optimization of left-right symmetry [8,10]. Subsequently, gait quality was estimated as median values over each week of 40 characteristics, reflecting walking speed, stride frequency, regularity, intensity, symmetry, smoothness, stability, and complexity of gait [8,10]. We report on 18 gait characteristics, that were significantly associated with prospective falls [10] in our tables, and provide data for all gait characteristics in Appendix A. We calculated a single gait quality composite score, based on a weighted sum of autocorrelation at stride frequency, power at step frequency, root mean square of the accelerations and index of harmonicity, which was previously found to be an important predictor for falls [10]. The custom Matlab-code to calculate this gait quality composite score can be found here: github.com/KimvanS/EstimateGaitQualityComposite.

### 2.4. Statistics

Data was inspected for normality using Q-Q plots and KS tests. Most variables followed a normal distribution, except for the amplitude of the dominant frequency. However, since its difference scores did follow a normal distribution, parametric statistics are reported for all variables. We tested for structural differences between both measurement weeks, using repeated measures ANOVAs, and Pearson correlations. The repeated measures ANOVAs were subsequently used to extract the overall mean and variance components, reflecting between-subjects (i.e., among participants), within-subject (i.e., between measurement weeks), and error variability (i.e., individual differences due to sampling error). Sample size calculations, for repeated measures, were performed to determine the number of participants required to detect changes in these gait quality characteristics over measurements. The number of participants (*n*) was estimated following [20] as: (1)$$n=\frac{2*s_{S}^{2}*\left(1-\left(r*\frac{s_{\mathrm{BS}}^{2}}{s_{S}^{2}}\right)\right)*{\left(t_{n-1,1-\beta }+t_{n-1,1-\alpha /2}\right)}^{2}}{\Delta^{2}}$$
where $s_{S}^{2}$ is the total variance, $s_{\mathrm{BS}}^{2}$ the between-subject variance, *r* the within-subject correlation, *t_df,p_* the *p*^th^-percentile of a t-distribution with *df* degrees of freedom, *1- β* the desired level of statistical power, *α* the desired level of statistical significance, and Δ the effect size. Statistical power was set to 0.8 and statistical significance was set to 0.05. The numbers of participants required to detect an effect size Δ of Cohen’s d 0.3 (small), 0.5 (medium), and 0.8 (large), were estimated for all gait quality characteristics.

## 3. Results

None of the gait quality characteristics differed significantly between the two measurement weeks (all p ≥ 0.11) and all were strongly correlated (r = 0.80 to 0.96; Figure 1 and Table 1). Between-subject variance was the largest variance component for all characteristics, and 3 to 26 times higher than within-subject variance. Moreover, between-subject variance components were highest for characteristics that were orientation invariant (e.g. walking speed, step length, and time), and generally higher for ML compared to AP, with VT in between (Table 1 and Table A1 for all 40 characteristics). 

The number of participants required to detect intervention effects with sufficient statistical power ranged from 8 (large effect, r = 0.9) to 343 (small effect, r = 0.3) (Figure 2 and Table 2; see Table A2 for all characteristics).

## 4. Discussion

The purpose of this paper was to report the variance components required for sample size calculations in studies focusing on characteristics of daily-life gait quality as outcome measures. This paper also aimed to determine the number of participants required to detect intervention effects in a repeated measure design with sufficient statistical power (here set at 80%). Our sample size calculations indicate that the number of participants required for a test-retest intervention design will range from 8 to 343, depending on the anticipated between-measurement correlation and effect size. These numbers seem reasonable, compared to the 200 to 500 (range 10 to 9940, median 230 [1]) participants that are generally included in fall prevention intervention studies. This approach has the additional advantage that outcomes are available directly after the intervention, and that physical activity or exposure, can be readily assessed using similar methods. Future studies are required to determine whether changes due to interventions focusing on improving balance and mobility, and reducing falls can indeed be detected with daily-life gait quality characteristics.

The high correlation between measurements, combined with the relatively small within-subject variance, suggests that participant’s daily-life gait characteristics are relatively stable, at least over weeks without an intervention. Possible changes in orientation, due to reattaching the sensor after water activities, may have increased within-subject variance, a source of variance that can be remediated by fixing a waterproof sensor directly onto the body. Changes in orientation, over days, may have contributed to the higher number of participants required for orientation-dependent characteristics, when compared to orientation-invariant characteristics. The relatively small within-subject variance is encouraging for clinical use, as it holds promise for evaluation on the individual level. 

The magnitude of the between-subject variance components for gait stability and variability characteristics in this study, was larger than those reported by Toebes and colleagues for treadmill gait [20]. This could be a result of differences in experimental setup and algorithms [21] to assess these characteristics. However, it is also possible that assessments in daily life result in better differentiation between individuals. A direct comparison of the sensitivity to change of the laboratory-based and daily-life assessment of gait quality characteristics for fall risk seems warranted.

Despite the benefits of using daily-life gait characteristics as outcomes in fall prevention trials, there are barriers to their broad uptake. Although, inertial sensors are relatively cheap compared to other motion registration equipment, they generally cost ~$500–5000. Moreover, these sensors require charging and setting-up, can be logistically inconvenient and need specialist code to extract meaningful outcomes. Finally, non-wearing might render collected data useless. Nevertheless, we feel that the advantages outweigh the disadvantages and hope that this paper provides the tools and data necessary to facilitate their implementation. 

This study has some limitations. The locomotion detection algorithm may have introduced some error. However, its validity and reliability are high [18,19]. Moreover, we limited this study to median values of the distribution of the daily-life gait quality characteristics while more extreme percentiles, that better reflect participants’ capacity [6,8], may show different variances. The estimated variance components will also depend on sensor characteristics, the methods of fixation and fixation location, and thus might not generalize to other settings. The strength of the correlation between measurements, used in the sample size calculations, was estimated and actual values may be lower, with longer follow-up times and individuals responding differently to interventions. The time between measurements was relatively short in this study, which warrants the assumption of little changes within individuals over time; longer follow-up periods in trials may therefore present larger within-subject variance. 

## 5. Conclusions

We provided variance estimates to determine the number of participants required for intervention trials using daily-life gait characteristics as outcomes. The subsequent sample size estimations indicate that a substantial number of participants is required to detect the effects of interventions. However, this number is lower than that commonly used in trials with incidence of falls as the outcome. Therefore, characteristics of daily-life gait quality seems to be a promising outcome for intervention studies, focusing on balance, mobility, and falls. 

## Figures and Tables

**Figure 1 sensors-19-04388-f001:**
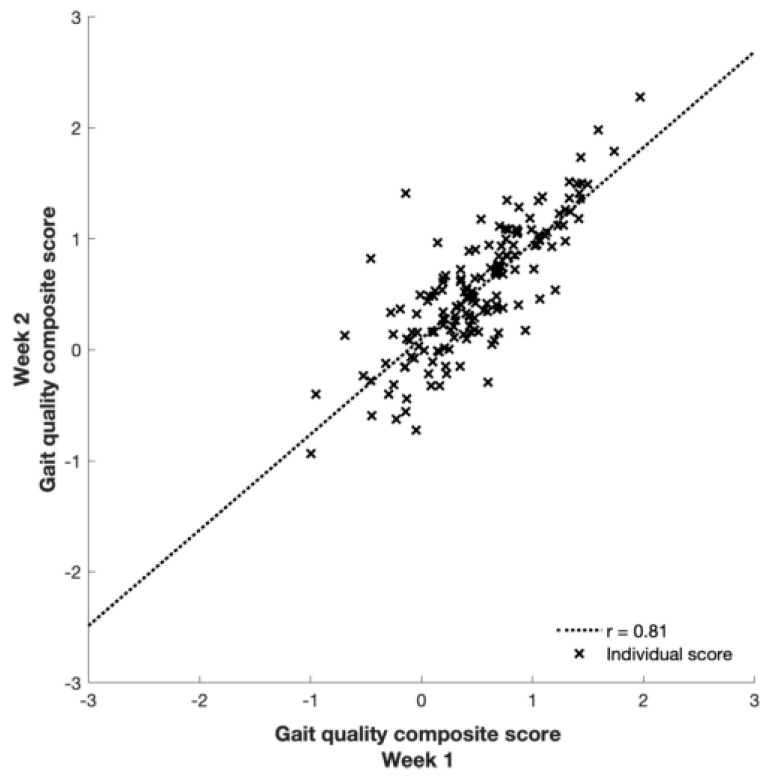
Agreement between the gait quality composite scores of the first and second week.

**Figure 2 sensors-19-04388-f002:**
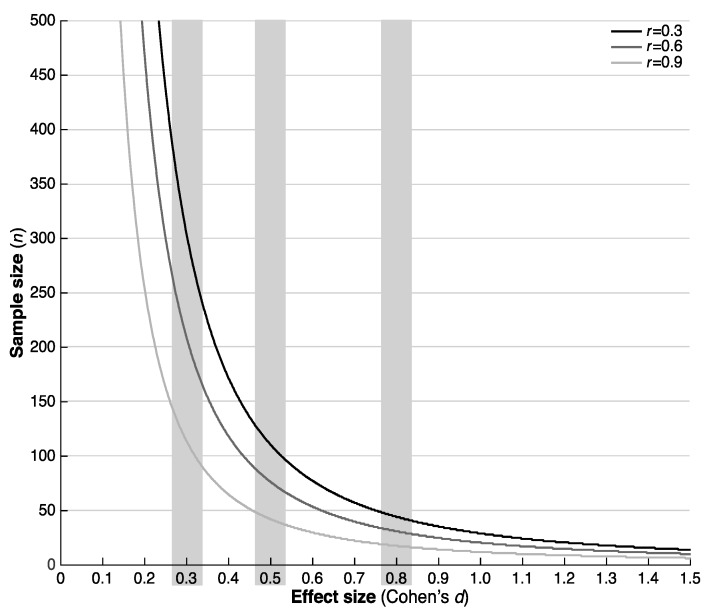
Number of participants required to detect an effect of Cohen’s *d* on the gait quality composite score.

**Table 1 sensors-19-04388-t001:** Agreement between weeks and estimated variance components for gait characteristics associated with prospective falls in [10].

	*p*	*r*	Mean	s^2^_BS_	s^2^_WS_	s^2^_E_
**Gait quality composite score**	0.64	0.81	0.51369	0.54700	0.08068	0.05759
**Walking speed**	0.39	0.93	0.45073	0.01030	0.00027	0.00036
**Stride frequency**	0.51	0.95	0.82482	0.01119	0.00013	0.00030
**Standard deviation VT**	0.77	0.92	1.32658	0.17858	0.00061	0.00733
**Standard deviation ML**	0.18	0.94	1.16407	0.09332	0.00560	0.00305
**Range AP**	0.40	0.91	7.33318	7.13497	0.23150	0.32894
**Stride autocorrelation VT**	0.54	0.82	0.37464	0.01051	0.00040	0.00103
**Stride autocorrelation AP**	0.63	0.83	0.32881	0.00945	0.00021	0.00088
**Amplitude of dominant frequency VT**	0.68	0.93	0.49412	0.04264	0.00026	0.00155
**Amplitude of dominant frequency ML**	1.00	0.91	0.47481	0.05827	9.26 × 10^−9^	0.00263
**Amplitude of dominant frequency AP**	0.11	0.85	0.51044	0.02357	0.00501	0.00198
**Width of dominant frequency AP**	0.14	0.80	0.75841	0.00611	0.00167	0.00075
**Index of harmonicity VT**	0.19	0.96	0.62043	0.04940	0.00156	0.00092
**Index of harmonicity ML**	0.68	0.92	0.64420	0.06977	0.00048	0.00276
**Harmonic ratio VT**	0.47	0.90	1.53576	0.07348	0.00211	0.00406
**Local divergence rate/stride VT**	0.75	0.87	2.11436	0.13693	0.00097	0.00919
**Local divergence rate/stride AP**	0.58	0.86	2.19507	0.09652	0.00217	0.00707
**Sample entropy ML**	0.74	0.91	0.31207	0.00394	2.07 × 10^−5^	0.00018

Note: We tested for structural differences between both measurement weeks using a repeated measures ANOVA (p-value) and Pearson correlations (r). We subsequently report mean values (mean) and variance components between-subjects (s^2^_BS_), within-subjects (s^2^_WS_) and due to sampling error (s^2^_E_).

**Table 2 sensors-19-04388-t002:** Number of participants required to detect an effect of Cohen’s *d* on gait characteristics associated with prospective falls in [10].

	Small Effect			Medium Effect			Large Effect		
	Cohen’s d = 0.3			Cohen’s d = 0.5			Cohen’s d = 0.8		
	r = 0.3	r = 0.6	r = 0.9	r = 0.3	r = 0.6	r = 0.9	r = 0.3	r = 0.6	r = 0.9
**Gait quality composite score**	303	208	114	110	76	42	44	31	18
**Walking speed**	259	158	56	95	58	22	38	24	10
**Stride frequency**	254	151	49	93	56	19	37	23	9
**Stride length**	251	148	44	92	54	17	37	22	8
**Standard deviation VT**	252	151	51	92	56	19	37	23	9
**Standard deviation AP**	266	163	61	97	60	23	39	25	10
**Range AP**	262	162	62	96	60	23	39	24	10
**Stride autocorrelation VT**	268	173	77	98	63	29	39	26	13
**Stride autocorrelation AP**	263	167	71	96	61	27	39	25	12
**Amplitude of dominant frequency VT**	253	151	50	92	56	19	37	23	9
**Amplitude of dominant frequency ML**	251	151	51	92	56	19	37	23	9
**Amplitude of dominant frequency AP**	323	227	130	118	83	48	47	34	20
**Width of dominant frequency AP**	343	250	156	125	91	57	50	37	24
**Index of harmonicity VT**	260	157	54	95	58	21	38	24	9
**Index of harmonicity ML**	253	152	51	92	56	20	37	23	9
**Harmonic ratio VT**	262	162	63	96	60	24	39	25	11
**Local divergence rate/stride VT**	256	157	59	93	58	23	38	24	10
**Local divergence rate/stride AP**	261	164	66	95	60	25	38	25	11
**Sample entropy ML**	253	153	53	92	56	20	37	23	9

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
