# Peer review of "Quality of Daily-Life Gait: Novel Outcome for Trials that Focus on Balance, Mobility, and Falls"

_sensors, 2019, doi:10.3390/s19204388_

Round 1

Reviewer 1 Report

1.    There are many other papers that worked on a same issue and may related to this work; however, those are not mentioned. As an instance:

Clemson, L., Singh, M. A. F., Bundy, A., Cumming, R. G., Manollaras, K., O’Loughlin, P., & Black, D. (2012). Integration of balance and strength training into daily life activity to reduce rate of falls in older people (the LiFE study): randomised parallel trial. Bmj345, e4547.

It would be great, if authors have more elaboration and comparison in the introduction and discussion, regarding to the literature review.

References are not sufficient.

2.    Since, the statistically significant might differ from real world significant, how can authors explain about using 80% sensitivity (Power) as significant?

3.    What was the probability distribution before using ANVOA?

4.    Just for better understanding, it would be great if authors provide an image from one of the participants who wore a trunk accelerometer.

Reviewer 2 Report

This manuscript has a clear purpose: To establish sample sizes for future studies were measures of daily-life gait characteristics are used, based on measurements from a single accelerometer worn at the waist. The method used is test-retest (one-week measurements spaced two weeks apart) with a population of 163 older people. Gait characteristics are calculated from the accelerometer measurements, and a particular  "gait quality composite score" (earlier publication by the  authors). Standard, well-motivated statistical analysis is performed on the repeated measurements in order to establish correlation and variances within-subject and between-subject, and using this to calculate required sample sized for different effect sizes. 

The article is well written, concise and easy to follow. The methodology is sound, and I appreciate very much that the matlab code for calculating the results is provided (github repository). The paper serves it purpose to guide further studies related to balance intervention and fall-risk, using gait characteristics in daily-life as proxy measures.

The only major concern I have with the paper is that its focus is not on the sensor technology, but rather on statistical analysis of results obtained using it. And therefore it is not in the main focus of the journal.  

Reviewer 3 Report

This manuscript presents a sample size and statistical power calculation for studies using daily-life gait quality as outcome. The paper is well written, and the data is clearly presented. However, some biostatistics journals might be a better fit to this manuscript than Sensors. Also, there are some major and minor concerns that need to be addressed before it can be published in a journal.

Major concerns:

The conclusion “quality of daily-life gait is a promising outcome for intervention studies focusing on improving balance and mobility and reducing falls” is lack of supportive materials. The entire manuscript is nothing more than sample size and variance analysis. It is a common sense, reported by many publications, that between-subject variance is larger within-subject variance. It cannot directly support the conclusion. It is a redundancy to reanalyze data in [10], and the data from the 163 people. Either dataset is good enough to do the job. The results and discussions failed to go through all the factors which may affect the sample size calculation, including some very important factors, such as gender, wearable sensor characteristics and locations.

Minor concerns:

The github link is not working. Grammar mistakes need to be corrected from line 98 to 100. Table I and Fig. 1 should be moved to the appendix as it is very straight forwarded.

Round 2

Reviewer 1 Report

The authors address all my concerns. 

Reviewer 3 Report

Thanks for addressing the comments!

It is a little weird to insert a subject image into a correlation plot in figure 1.
